# $S^2$**FGL: Spatial Spectral Federated Graph Learning**

**Zihan Tan** [* 1]   **Suyuan Huang** [* 1]   **Guancheng Wan** [1]   **Wenke Huang** [1]   **He Li** [1]   **Mang Ye** [1]

## Abstract

Federated Graph Learning (FGL) combines the privacy-preserving capabilities of Federated Learning (FL) with the strong graph modeling capability of Graph Neural Networks (GNNs). Current research addresses subgraph-FL from the structural perspective, neglecting the propagation of graph signals on the spatial and spectral domains of the structure. From a spatial perspective, subgraph-FL introduces edge disconnections between clients, leading to disruptions in label signals and a degradation in the semantic knowledge of the global GNN. From a spectral perspective, spectral heterogeneity causes inconsistencies in signal frequencies across subgraphs, which makes local GNNs overfit the local signal propagation schemes. As a result, spectral client drift occurs, undermining global generalizability. To tackle the challenges, we propose a global knowledge repository to mitigate the challenge of poor semantic knowledge caused by label signal disruption. Furthermore, we design a frequency alignment to address spectral client drift. The combination of **S**patial and **S**pectral strategies forms our framework $S^2$FGL. Extensive experiments on multiple datasets demonstrate the superiority of $S^2$FGL. The code is available at https://github.com/Wonder7racer/S2FGL.git

## 1. Introduction

Graph Neural Networks (GNNs) have demonstrated remarkable efficacy in modeling graph-structured data (Wan et al., 2025a; Fang et al., 2025), thereby finding applications across various domains, such as social networks (Fan et al., 2020; Zhang et al., 2022b), epidemiology (Liu et al., 2024), and

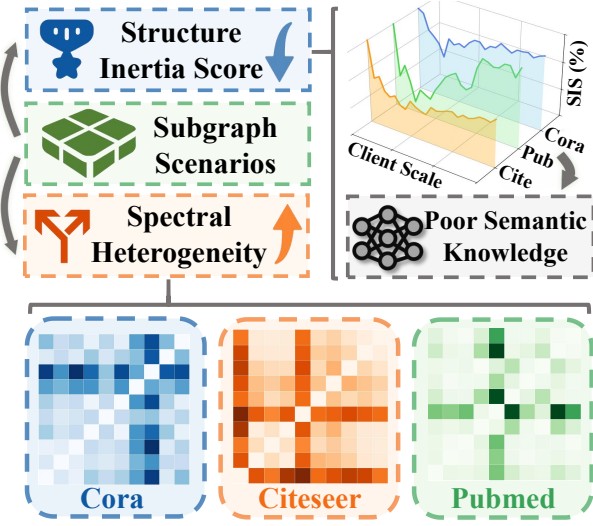

*Figure 1.* In the first place, compared with centralized GNN training, subgraph-FL is encountering **label signal disruption** challenge, leading to decreased structure inertia score and **poor semantic knowledge** for GNNs. Moreover, we demonstrate the heat map of the Kullback-Leibler divergence of eigenvalue distributions across clients. Inconsistency in subgraph signal frequency caused by **spectral heterogeneity** leads to **spectral client drift**.

fraud detection (Wang et al., 2019; Tang et al., 2022). However, in real-world scenarios, graph data is often generated at the edge devices rather than in centralized systems (Zhang et al., 2021a). To address this, Federated Graph Learning (FGL) has emerged (Fu et al., 2022; Liu & Yu, 2022; Tan et al., 2025; 2024; Huang et al., 2022; Wan et al., 2024b; 2025b), leveraging the data privacy-preserving capabilities of Federated Learning (FL) (Huang et al., 2024; 2023b;c; 2022) to enable the efficient distributed training of GNNs (Huang et al., 2024). A prominent application of FGL is subgraph-FL, in which each participant holds a subgraph derived from the same overall graph data.

Although numerous FGL methods have attempted to provide solutions based on structure to enhance effectiveness, including identifying structurally similar collaborators (Baek et al., 2023; Xie et al., 2021; Li et al., 2024), enhancing structural knowledge exchange (Tan et al., 2023; Huang et al., 2023a; Tan et al., 2025), and retrieving generic information under structural shifts (Wan et al., 2024a; Tan et al., 2024). Nevertheless, these approaches overlooked the propagation

---

*Equal contribution [1]National Engineering Research Center for Multimedia Software, School of Computer Science, Wuhan University, Wuhan, China. Correspondence to: Mang Ye <yemang@whu.edu.cn>.

*Proceedings of the $42^{nd}$ International Conference on Machine Learning*, Vancouver, Canada. PMLR 267, 2025. Copyright 2025 by the author(s).

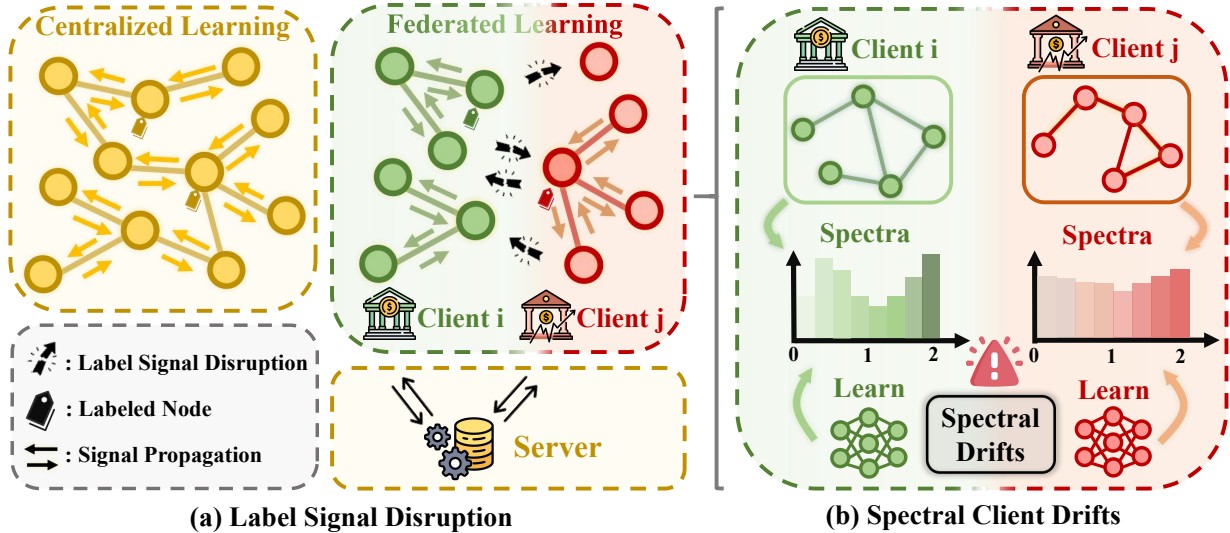

*Figure 2.* **Problem Illustration.** (a) From the spatial perspective, nodes in subgraph-FL lose label signals from originally nearby labeled nodes due to edge loss, namely **label signal disruption**. Correspondingly, GNNs suffer from **poor semantic knowledge**, leading to a deteriorated global GNN. (b) From the spectral perspective, **spectral heterogeneity** induces inconsistencies in signal frequencies across subgraphs, leading to **spectral client drift** in the signal propagation paradigms of GNNs and degraded global generalizability.

of graph signals within the structure. Specifically, graph signal propagation can be analyzed from two perspectives: the spatial and the spectral domain. Specifically, the spatial domain governs the explicit transmission of signals among linked nodes, while the spectral domain characterizes signal diffusion across varying frequency spectra.

From the spatial perspective, due to edge loss, we hypothesize that nodes in subgraph-FL lose label signals from originally nearby labeled nodes. This degradation hampers the ability of GNNs to learn comprehensive semantic knowledge, resulting in poor global performance and reduced generalizability. Correspondingly, we define this phenomenon as label signal disruption, which naturally exists in subgraph-FL. For verifying, inspired by graph active learning research (Han et al., 2023), we investigate how the Structure Inertia Score (SIS) varies in subgraph-FL. Specifically, SIS evaluates the influence and significance of labels on graphs. In Fig. 1, we empirically demonstrate that the SIS decreases in subgraph-FL compared with centralized training. Correspondingly, existing methods suffer from poor semantic knowledge. Based on our empirical analysis, we pose the question: **I)** *How can we address the challenge of poor semantic knowledge under label signal disruption?*

From the spectral perspective, inconsistencies in signal frequencies across clients caused by spectral heterogeneity induce spectral client drift in the signal transmission schemes of GNNs, thereby undermining the collaboration. To verify this phenomenon, we examine graph spectra across clients and demonstrate the heterogeneity in Fig. 1. It reveals inconsistent eigenvalue distribution across clients. As a result, GNNs learn distinct signal propagation schemes of sub-

graphs and optimize in different spectral directions, leading to **spectral client drift** and degraded generalizability. Based on our analysis, we pose the question: **II)** *How can we alleviate spectral client drift under spectral heterogeneity?*

To address the challenge of poor semantic knowledge under label signal disruption in Question **I)**, we propose Node Label Information Reinforcement (NLIR). Specifically, our strategy leverages structurally and semantically representative nodes to construct a prototype-based global repository of semantic knowledge. During training, NLIR calculates the similarity distribution between all representative prototypes with node features, which provides multidimensional semantic localization of nodes. Consequently, our strategy injects semantic knowledge from the repository into the local GNN during training, effectively mitigating the issue of poor semantic knowledge under label signal disruption.

Considering the spectral client drift posed by spectral heterogeneity in **II)**, we propose Frequency-aware Graph Modeling Alignment (FGMA). Our method utilizes the similarity relationship of the node feature of the frozen global GNN and the local GNN to reconstruct spectra that incorporates GNNs adjacency awareness. FGMA then projects the high-frequency and low-frequency components of the features onto this spectrum. Subsequently, by aligning the local projections with the global one, we encourage the GNNs to learn a globally generic frequency processing scheme, thereby mitigating spectral client drift.

In conclusion, our key contributions are:

- First, we identify and empirically reveal the issue of

poor semantic knowledge under label signal disruption. In addition, we reveal the spectral client drift under spectral heterogeneity in subgraph-FL.

- We design our framework $S^2$FGL including strategy Node Label Information Reinforcement and Frequency-aware Graph Modeling Alignment, effectively addressing the challenges of poor semantic knowledge and spectral client drift in subgraph-FL.

- We conduct extensive experiments on various datasets, validating the superiority of our proposed $S^2$FGL.

## 2. Related Work

**Federated Graph Learning.** Federated graph learning leverages the powerful graph modeling capabilities of GNNs along with the privacy-preserving attributes of federated learning, thus gaining increasing attention these days (He et al., 2021a; Fu et al., 2022; Liu & Yu, 2022; Wan et al., 2025b). Current FGL research can generally be categorized into two types: intra-graph FGL and inter-graph FGL. Intra-graph FGL research primarily focuses on subgraph-FL scenarios, where each client participates in the collaboration with a part of the whole graph (Zhang et al., 2021b). Correspondingly, the training targets include missing link prediction (Chen et al., 2021; Baek et al., 2023), node classification (Huang et al., 2023a; Li et al., 2024; Wan et al., 2024a; Zhu et al., 2024), and so on. On the other hand, clients in inter-graph FGL own independent local graph data, such as multiple graphs from different domains (Tan et al., 2023; Xie et al., 2021). In this paper, we focus on subgraph-FL scenarios of intra-graph FGL. Specifically, we are the first to empirically reveal and address the challenge of poor semantic knowledge under label signal disruption and client drift under spectral heterogeneity among subgraphs, while existing methods inevitably fail spatially and spectrally due to the lack of targeted solutions.

**Federated Learning.** Federated learning (Huang et al., 2023c; 2024; Yang et al., 2023; Wan et al., 2024a) has gained increasing attention in recent years as it addresses the issue of data silos while ensuring data privacy. Several research directions have emerged from FL, including robustness (Xu et al., 2022; Hong et al., 2023; Zhu et al., 2023; Fang & Ye, 2022), fairness (Chen et al., 2024; Ezzeldin et al., 2023; Ray Chaudhury et al., 2022), and asynchronous federated learning (Xu et al., 2023; Zhang et al., 2023d). Generally, FL can be categorized into two main types by their optimization objective: traditional FL (tFL) and personalized FL (Hu et al., 2024; Shang et al., 2022; Lv et al., 2024; Smith et al., 2017). Research of tFL aims at aggregating a highly generalizable global model (McMahan et al., 2017; Li et al., 2020; Acar et al., 2021; Zhang et al., 2022a). For instance, FedNTD (Lee et al., 2022) preserves the global

perspective on local data for the not-true classes, FEDGEN (Zhu et al., 2021) ensembles user information in a data-free manner to regulate local training, and SCAFFOLD (Karimireddy et al., 2020) uses variance reduction for the client drift phenomenon. Instead, strategies of personalized FL (pFL) aim to customize models that perform optimally for each client (Wu et al., 2023; Zhou & Konukoglu, 2023; Li et al., 2021; Zhang et al., 2023b). Specifically, FedALA (Zhang et al., 2023c) proposed adaptive masks to achieve personalized aggregation, DBE (Zhang et al., 2023a) stores domain biases for elimination, and FedRoD (Chen & Chao, 2022) leverages two heads for global and personalized tasks.

**Graph Spectrum** Being related closely to graph connectivity, signal propagation, and structure, graph spectra have proven essential in performing various tasks on graph-structured data. For instance, it plays an essential role in anatomy detection, (Gao et al., 2023; Tang et al., 2022), graph condensation (Kreuzer et al., 2021; Liu et al., 2023), and graph contrastive learning (Bo et al., 2023a; Liu et al., 2022). Additionally, spectral GNNs (Wu et al., 2020) based on spectral filters are showing powerful ability in modeling graph data and attracting more attention. Specifically, existing research either (He et al., 2021b; Defferrard et al., 2016; He et al., 2022; Wang & Zhang, 2023) leverages various orthogonal polynomials to approximate arbitrary filters, or utilizes neural networks to parameterize the filters (Liao et al., 2019; Bo et al., 2023b). Although the potential of graph spectrum has been explored in various scenarios and tasks, the spectral domain in generalizable subgraph-FL has remained unexplored. Consequently, current methods suffer from optimization diverging on spectra and are trapped in suboptimal learning. Instead, our approach remarkably mitigates the challenge by targeted alignment on spectra.

> **Graph Signal Propagation:** *Graph signal propagation describes how node signals diffuse on graph structures. In the* **spatial** *domain, propagation occurs through explicit signal passing along edges. In the* **spectral** *domain, propagation is characterized by how signals distribute across different frequency components.* **Label Signal Disruption:** *As subgraphs experience edge loss, nodes lose critical label signals containing class knowledge from their formerly adjacent labeled neighbors. Consequently, it limits the ability of GNNs to capture class distinctions accurately, leading to poor semantic knowledge under label signal disruption.* **Spectral Client Drift:** *Inconsistencies in signal frequencies on graph spectra across subgraphs lead to spectral heterogeneity and diverging signal propagation schemes, causing spectral client drift and degrading the generalizability of the global model.*

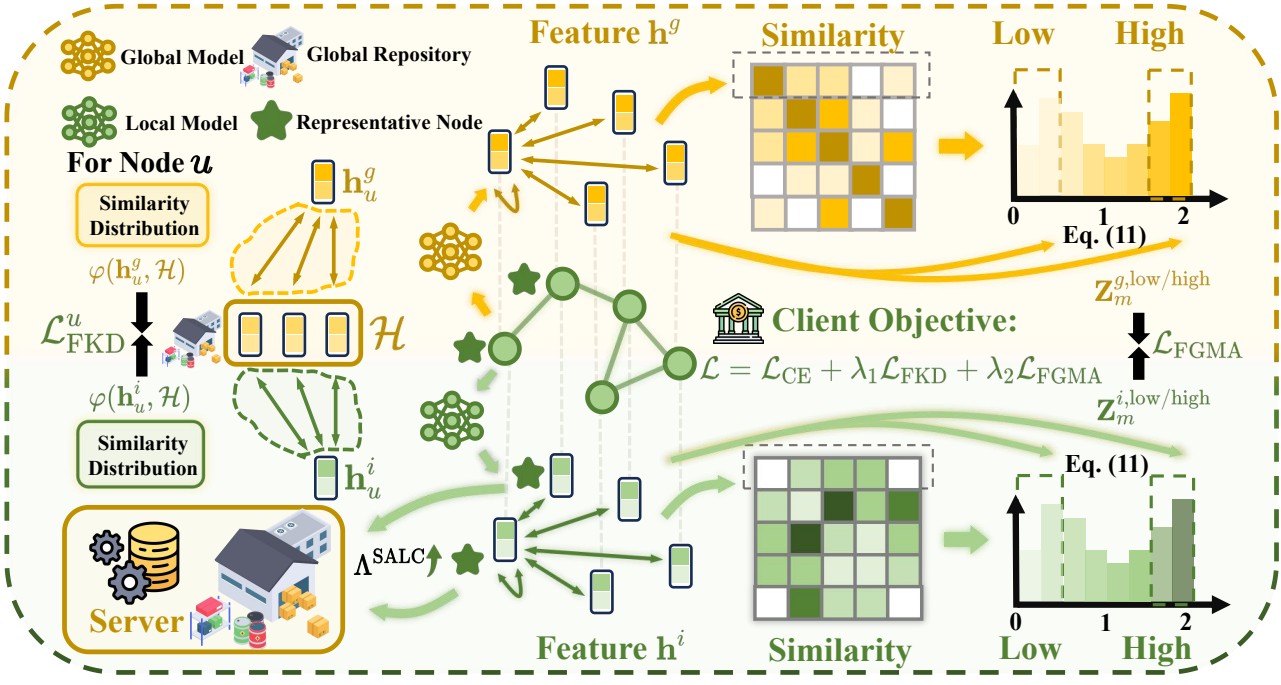

(a) **Node Label Information Reinforcement**          (b) **Frequency-aware Graph Modeling Alignment**

*Figure 3.* **Framework Illustration.** (a) Node Label Information Reinforcement (NLIR) leverages a structurally and semantically representative global prototype repository. It provides **multidimensional semantic localization** of nodes through similarity distribution and allows $L_{\text{FKD}}$ to **inject** the semantic knowledge during training. (b) Frequency-aware Graph Modeling Alignment (FGMA) **aligns** local high and low **spectral adjacency awareness** with the global GNN for a generic signal propagation scheme, mitigating spectral drifts.

## 3. Problem Statement

**Notation**. Let the graph data be represented as $\mathcal{G} = (\mathcal{V}, \mathcal{E})$, where $\mathcal{V}$ is the set of nodes with $|\mathcal{V}| = N$ vertices, and $\mathcal{E} \subseteq \mathcal{V} \times \mathcal{V}$ denotes the set of edges connecting these nodes. The adjacency matrix is represented by $\mathbf{A} \in \mathbb{R}^{N \times N}$, where $\mathbf{A}_{uv} = 1$ indicates the presence of an edge $e_{uv} \in \mathcal{E}$, and $\mathbf{A}_{uv} = 0$ otherwise. Moreover, $\mathbf{X}$ the feature vector matrix of the graph $\mathcal{G}$. The Laplacian matrix is given by $\mathbf{L} = \mathbf{D} - \mathbf{A}$, where $\mathbf{D}$ is the degree matrix. The unitary matrix $\mathbf{U}$ is composed of the eigenvectors of $\mathbf{L}$. To distinguish between local and global properties, we introduce the following notation: the symbol $i$ represents local properties or entities, whereas $g$ denotes global properties or entities.

**Definition 3.1. Personalized PageRank (PPR)**: *The PPR matrix quantifies the influence each node has on every other node within the graph and is defined as:*

$$P = \alpha \left( I - (1 - \alpha) D^{-1} A \right)^{-1}. \tag{1}$$

*Here, $\alpha \in (0, 1)$ is the teleportation probability, representing the probability of the random walk restarting from the source node. A typical value is $0.15$, which makes the continuation probability $(1 - \alpha) = 0.85$. $I$ is the identity matrix, $A$ is the adjacency matrix of the graph, and $D$ is the degree matrix with $D_{ii}$ denoting the degree of node $i$.*

**Definition 3.2. Structure Inertia Score (SIS)**: *The SIS quantifies the cumulative influence of the training nodes on the entire graph and is defined as:*

$$SIS(P, t) = \sum_{i=1}^{n} \max_{j} \left( P_{i,j} \cdot t_j \right). \tag{2}$$

*Here, $P$ is the PPR matrix, and $t \in \{0, 1\}^n$ is a binary vector indicating the training nodes, where $t_j = 1$ if node $j$ is part of the training set, and $t_j = 0$ otherwise. The SIS aggregates the maximum personalized PageRank values from each node to any labeled node, effectively measuring the strongest influence each node in the graph receives from the training set. A higher SIS indicates greater structural inertia, suggesting that the labels have a significant influence over the network overall graph structure.*

## 4. Methodology

### 4.1. Motivation

Signal propagation over graph structures fundamentally shapes the signal transmission paradigm of GNNs. Therefore, rather than focusing solely on challenges arising from static graph structures in subgraph-FL, it is crucial to consider the dynamics of signal propagation. Correspondingly,

we conduct our analysis from both the spatial and spectral domains from the perspective of the graph signal. Specifically, the spatial domain governs the explicit passing of signals between connected nodes, while the spectral domain captures signal diffusion across different frequencies. Accordingly, we empirically validate the presence of two major challenges from the spatial and spectral perspectives: **label signal disruption** and **spectral client drift**. These phenomena respectively pose challenges of **poor semantic knowledge** and **spectral client drift**, which severely constrain the potential of subgraph-FL collaboration.

**Motivation of NLIR.** Graph data is fragmented across clients in subgraph-FL, which inevitably disrupts semantic signals from labeled nodes across clients. Label signal disruption undermines key pathways for propagating semantic information. This results in biased local feature representations, which ultimately degrade the performance of GNNs. For validation, we investigate the relationship between the decrease in SIS and the client scale. Specifically, the SIS exhibits a downward trend as the client scale increases, with notably lower scores in the range emphasized by mainstream subgraph-FL studies, thereby highlighting the label signal disruption phenomenon. We aim to mitigate its negative impact by preserving valuable semantic knowledge across fragmented subgraphs. Correspondingly, we propose NLIR. By selecting nodes with both structural representativeness and rich semantic information for the construction of a global repository and injecting it during local training, NLIR reduces the information loss inherent in the subgraph scenarios. Subsequently, NLIR assesses the similarity distributions between node features and all representative prototypes for both local and global GNNs during local training, thereby enabling multidimensional semantic localization of nodes. By aligning the two similarity distributions, it effectively injects semantic knowledge and enhances feature modeling semantically.

**Motivation of FGMA.** We reveal the challenge of spectral client drift in subgraph-FL in Fig. 1, where GNNs across different clients capture frequency information inconsistently due to graph spectral heterogeneity. This further leads to overfitting local signal propagation frequencies, which brings spectral conflicts during collaboration and compromises the generalizability of the global GNN. To address spectral client drift, we propose the Frequency-aware Graph Modeling Alignment. Specifically, FGMA reconstructs the local graph spectra with the GNN adjacency awareness by calculating the node similarity matrix. Subsequently, by projecting feature representations separately onto high- and low-frequency components of the reconstructed spectra and aligning the local and global projections, FGMA promotes the learning of a generalizable spectral signal propagation paradigm across clients, thereby reducing frequency-based discrepancies during collaboration and mitigating spectral

client drift. Consequently, our strategy effectively enhances the global generalizability spectrally.

### 4.2. Node Label Information Reinforcement

First of all, we introduce the Structure-Aware Label Centrality (SALC) metric, denoted as $\Lambda_u^{\text{SALC}}$ for node $u$. It is defined as the combination of the label influence centrality $\Lambda_u^l$ and the structural prominence score $\Lambda_u^s$:

$$\Lambda_u^{\text{SALC}} = \Lambda_u^s + \Lambda_u^l, \tag{3}$$

where $\Lambda_u^s$ assesses the structural representativeness of node $u$, while $\Lambda_u^l$ quantifies the influence propagation of labels.

$$\Lambda_u^s = \max\left(\tilde{P}_{u,v} \cdot \tau_v\right), \quad \Lambda_u^l = \sum_{v \in \mathcal{V}_L} \tilde{P}_{v,u}^{(L)}, \tag{4}$$

where $\tilde{P}_{u,v}$ is the $(u,v)$-th element of the standard PPR matrix $\tilde{\mathbf{P}}$, and $\tau_v$ represents the prior importance score of node $v$, typically initialized to 1 for all nodes. The structural prominence score captures the maximum influence exerted by any node on node $u$, weighted by its importance. As clarified, the PPR matrix used for computing label influence centrality is denoted as $\tilde{\mathbf{P}}^{(L)}$, and defined as:

$$\tilde{\mathbf{P}}^{(L)} = \alpha\left(\mathbf{I} - (1-\alpha)\mathbf{D}^{-1}\mathbf{A}'\right)^{-1}. \tag{5}$$

Here, $\mathcal{V}_L$ denotes the set of labeled nodes, and $\tilde{P}_{v,u}^{(L)}$ represents the influence of node $v$ on node $u$ as captured by $\tilde{\mathbf{P}}^{(L)}$. To accurately capture the influence of labeled nodes, the inclusion of self-loops in $\mathbf{A}'$ ensures that each labeled node's own label contributes to its $\Lambda_u^l$. Compared to the intuitive approach of directly selecting labeled nodes, the SALC metric $\Lambda_u^{\text{SALC}}$ considers both structural representativeness of the nodes and diffusion of label signals, thus avoiding biases caused by isolated labeled nodes. It is also capable of selecting unlabeled nodes that still possess rich label signals and structural advantages. This improves knowledge quality, enriches the repository, and mitigates the label signal disruption problem. After computing the SALC scores, we rank the nodes based on their $\Lambda_u^{\text{SALC}}$ values and select the top $K$ nodes, where the default value of $K$ is $1/3$ of the total number of nodes. Subsequently, for each class $c$, the local prototype $\mathbf{H}_c^i$ at each client is computed as the mean feature vector of the selected nodes belonging to class $c$:

$$\mathbf{H}_c^i = \frac{1}{|\mathcal{V}_c^i|} \sum_{u \in \mathcal{V}_c^i} \mathbf{h}_u^i, \tag{6}$$

where $\mathcal{V}_c^i$ represents the set of selected nodes categorized as class $c$ on client $i$, and $\mathbf{h}_u^i$ is the feature vector of node $u$ on client $i$. Once the local prototypes are computed, clients upload their prototypes to the server along with the node count. For each class $c$, the server aggregates the

prototypes from $\alpha$ percent of the clients by weighting each local prototype according to its sample size. Four global anchor prototypes are constructed for each class. Each global prototype $\mathbf{H}_c^{g,k}$ is computed as:

$$\mathbf{H}_c^{g,k} = \frac{1}{\sum_{i \in \mathcal{N}_c^k} |\mathcal{V}_c^i|} \sum_{i \in \mathcal{N}_c^k} |\mathcal{V}_c^i| \mathbf{H}_c^i, \quad \mathcal{H} = \begin{bmatrix} \mathbf{H}_1^{g,1} \\ \mathbf{H}_1^{g,2} \\ \mathbf{H}_1^{g,3} \\ \vdots \\ \mathbf{H}_C^{g,4} \end{bmatrix}, \quad (7)$$

where $\mathbf{H}_c^{g,k}$ represents the $k$-th global prototype for class $c$, $\mathcal{N}_c^k$ is the set of clients randomly selected to contribute to the $k$-th global prototype for class $c$, and $C$ is the number of classes. The global repository $\mathcal{H}$ contains all the global prototypes and will be broadcast back to clients. After the global knowledge repository is constructed, it is distributed to the local clients along with the model parameters. The global features $\mathbf{h}_u^g$ used in the following loss formulation refer to the frozen inference features extracted locally using the distributed global model. To regulate local training, we propose a federated knowledge distillation loss function aimed at harmonizing the semantic feature localization of the local GNN with its global counterpart, namely by aligning their similarity distributions for the representative prototypes stored in the global repository:

$$\mathcal{L}_{\text{FKD}} = \frac{1}{|\mathcal{V}_i|} \sum_{u \in \mathcal{V}_i} \text{KL} \left( \sigma \left( \varphi(\mathbf{h}_u^i, \mathcal{H}) \right), \sigma \left( \varphi(\mathbf{h}_u^g, \mathcal{H}) \right) \right), \quad (8)$$

where $\text{KL}(\cdot, \cdot)$ represents the Kullback-Leibler divergence, and $\sigma(\cdot)$ is the softmax function applied to similarity scores computed by the function $\varphi(\mathbf{h}, \mathcal{H})$, which returns a vector of cosine similarities between the feature and all prototypes.

### 4.3. Frequency-aware Graph Modeling Alignment

To emphasize the GNN spectral adjacency awareness and more accurately capture similarities between nodes, we leverage the feature matrix $\mathbf{h}$ to compute node similarity matrices. In the construction of the following similarity matrix, $\mathbf{h}$ denotes operations applied to both $\mathbf{h}^i$ and $\mathbf{h}^g$. Specifically, for each node $u$, we identify its $k_{\text{sim}}$ most similar neighbors based on the cosine similarity of their feature vectors $\mathbf{h}_u$. We then construct a sparse self-similarity matrix $\mathbf{S}'$ as:

$$\mathbf{S}'_{u,v} = \begin{cases} \frac{\mathbf{h}_u \cdot \mathbf{h}_v}{\|\mathbf{h}_u\|_2 \|\mathbf{h}_v\|_2} & v \text{ among the top } k_{\text{sim}} \\ 0 & \text{otherwise} \end{cases}. \quad (9)$$

Subsequently, we calculate the graph laplacian matrix $\mathbf{L}'$ based on this sparse similarity matrix $\mathbf{S}'$ as:

$$\mathbf{L}' = \mathbf{D}' - \mathbf{S}', \quad (10)$$

where $\mathbf{D}'$ is the diagonal degree matrix of $\mathbf{S}'$. Moreover, we perform eigendecomposition on the laplacians $\mathbf{L}'^i$ and $\mathbf{L}'^g$.

For $\mathbf{L}'^i$, let $\{\mathbf{u}_m^{\text{low},i}\}_{m=1}^{k_{\text{eig}}}$ be the eigenvectors corresponding to the smallest eigenvalues, which represents low-frequency. While $\{\mathbf{u}_m^{\text{high},i}\}_{m=1}^{k_{\text{eig}}}$ denotes the largest eigenvalues, which represents high-frequency. Similarly, for $\mathbf{L}'^g$, we obtain $\{\mathbf{u}_m^{\text{low},g}\}_{m=1}^{k_{\text{eig}}}$ and $\{\mathbf{u}_m^{\text{high},g}\}_{m=1}^{k_{\text{eig}}}$. The feature matrix $\mathbf{h}$ is then projected onto each of these eigenvectors. For instance:

$$\mathbf{Z}_m^{\text{low}} = (\mathbf{u}_m^{\text{low}} \mathbf{u}_m^{\text{low}T}) \mathbf{h}, \quad \mathbf{Z}_m^{\text{high}} = (\mathbf{u}_m^{\text{high}} \mathbf{u}_m^{\text{high}T}) \mathbf{h}. \quad (11)$$

Applying these projections for each $m \in \{1, \dots, k_{\text{eig}}\}$, we obtain several sets of projected feature matrices used in the loss computation. Specifically, local features $\mathbf{h}^i$ are projected onto low/high-frequency eigenvectors of the local graph, yielding $\mathbf{Z}_m^{i,\text{low}}$ and $\mathbf{Z}_m^{i,\text{high}}$, respectively. Similarly, frozen global inference features $\mathbf{h}^g$ are projected onto corresponding eigenvectors from $\mathbf{L}'^g$, yielding $\mathbf{Z}_m^{g,\text{low}}$ and $\mathbf{Z}_m^{g,\text{high}}$. Conequently, loss $\mathcal{L}_{\text{FGMA}}$ is then defined as the sum of MSE over all eigenvector-projected pairs:

$$\mathcal{L}_{\text{FGMA}} = \sum_{m=1}^{k_{\text{eig}}} \left( \text{MSE}(\mathbf{Z}_m^{i,\text{low}}, \mathbf{Z}_m^{g,\text{low}}) + \text{MSE}(\mathbf{Z}_m^{i,\text{high}}, \mathbf{Z}_m^{g,\text{high}}) \right).$$
$$(12)$$

This loss addresses spectral heterogeneity by aligning client and global signal characteristics in both spectral domains. Combining strategies Node Label Information Reinforcement and Frequency-aware Graph Modeling Alignment, our framework $S^2$FGL reinforces semantic knowledge during local modeling and mitigates spectral client drift. Ultimately, the loss for local training is:

$$\mathcal{L} = \mathcal{L}_{\text{CE}} + \lambda_1 \mathcal{L}_{\text{FKD}} + \lambda_2 \mathcal{L}_{\text{FGMA}}, \quad (13)$$

where $\mathcal{L}_{\text{CE}}$ denotes the standard cross-entropy loss for node classification, while $\lambda_1$ and $\lambda_2$ are balancing hyperparameters for the proposed methods NLIR and FGMA.

## 5. Experiments

### 5.1. Experimental Setup

**Datasets.** We conducted experiments on various datasets to validate the superiority of our framework $S^2$FGL. The homophilic graph datasets include Cora, Citeseer, and Pubmed, while the heterophilic graph datasets comprise Texas, Wisconsin, and Minesweeper. The following provides a description of each dataset. **Cora** (McCallum et al., 2000) dataset consists of 2708 scientific publications classified into one of seven classes. There are 5429 edges in the network of citations. 1433 distinct words make up the dictionary. **Citeseer** (Giles et al., 1998) dataset consists of 3312 scientific publications classified into one of six classes and 4732 edges. The dictionary contains 3703 unique words. **Pubmed** (Sen et al., 2008) dataset consists of 19717 scientific papers on diabetes that have been categorized into one of three categories in the PubMed database. The citation network has 44338 edges.

*Table 1.* **Performance Comparison** with the state-of-the-art methods on homophilic and heterophilic graph datasets. We report the node classification accuracies with the performance improvement over FedAvg. The best results are highlighted in bold.

| Methods | Cora | CiteSeer | PubMed | Texas | Wisconsin | Minesweeper |
|---------|------|----------|--------|-------|-----------|-------------|
| FedAvg [ASTAT17] | $81.9 \pm 0.7$ | $74.3 \pm 0.4$ | $87.3 \pm 0.3$ | $72.8 \pm 2.2$ | $77.6 \pm 2.7$ | $79.6 \pm 0.1$ |
| FedProx [arXiv18] | $82.1 \pm 0.5$ ↑0.2 | $74.4 \pm 0.3$ ↑0.1 | $87.9 \pm 0.4$ ↑0.6 | $73.5 \pm 3.7$ ↑0.7 | $77.3 \pm 3.4$ ↓0.3 | $79.7 \pm 0.1$ ↑0.1 |
| FedNova [NeurIPS20] | $81.6 \pm 1.2$ ↓0.3 | $74.4 \pm 0.4$ ↑0.1 | $88.2 \pm 0.5$ ↑0.9 | $73.0 \pm 4.4$ ↑0.2 | $77.4 \pm 4.2$ ↓0.2 | $79.9 \pm 0.4$ ↑0.3 |
| FedFa [ICLR23] | $82.7 \pm 0.5$ ↑0.8 | $74.9 \pm 0.6$ ↑0.6 | $87.8 \pm 0.5$ ↑0.5 | $73.9 \pm 3.6$ ↑1.1 | $78.1 \pm 4.6$ ↑0.5 | $80.1 \pm 0.3$ ↑0.5 |
| FedSage+ [NeurIPS19] | $82.3 \pm 0.7$ ↑0.4 | $75.2 \pm 0.3$ ↑0.9 | $88.2 \pm 0.7$ ↑0.9 | $73.7 \pm 4.0$ ↑0.9 | $\mathbf{79.0 \pm 3.3}$ ↑1.4 | $79.9 \pm 0.2$ ↑0.3 |
| FedStar [AAAI23] | $82.6 \pm 0.5$ ↑0.7 | $74.5 \pm 0.3$ ↑0.2 | $88.1 \pm 0.6$ ↑0.8 | $74.3 \pm 2.7$ ↑1.5 | $78.3 \pm 4.7$ ↑0.7 | $79.8 \pm 0.1$ ↑0.2 |
| FedPub [ICML23] | $82.3 \pm 0.8$ ↑0.4 | $74.8 \pm 0.7$ ↑0.5 | $88.0 \pm 0.4$ ↑0.7 | $73.4 \pm 3.5$ ↑0.6 | $77.8 \pm 3.1$ ↑0.2 | $79.9 \pm 0.2$ ↑0.3 |
| FGSSL [IJCAI23] | $82.6 \pm 0.4$ ↑0.7 | $74.9 \pm 0.2$ ↑0.6 | $87.6 \pm 0.7$ ↑0.3 | $73.6 \pm 4.6$ ↑0.8 | $77.8 \pm 3.8$ ↑0.2 | $79.9 \pm 0.2$ ↑0.3 |
| FedGTA [VLDB24] | $82.4 \pm 0.8$ ↑0.5 | $75.1 \pm 0.5$ ↑0.8 | $87.7 \pm 0.9$ ↑0.4 | $72.6 \pm 4.2$ ↓0.2 | $77.8 \pm 4.1$ ↑0.2 | $80.2 \pm 0.3$ ↑0.6 |
| FGGP [AAAI24] | $82.5 \pm 0.4$ ↑0.6 | $74.7 \pm 0.5$ ↑0.4 | $87.5 \pm 0.4$ ↑0.2 | $73.6 \pm 2.8$ ↑0.8 | $78.2 \pm 3.4$ ↑0.6 | $80.4 \pm 0.3$ ↑0.8 |
| $S^2$**FGL (ours)** | $\mathbf{83.4 \pm 0.2}$ ↑1.5 | $\mathbf{76.0 \pm 0.3}$ ↑1.7 | $\mathbf{88.6 \pm 0.2}$ ↑1.3 | $\mathbf{74.8 \pm 2.3}$ ↑2.0 | $\mathbf{79.0 \pm 1.0}$ ↑1.4 | $\mathbf{80.5 \pm 0.1}$ ↑0.9 |

**Texas** and **Wisconsin** datasets are subsets of the WebKB dataset (Craven et al., 1998). The WebKB dataset was introduced in 1998, comprising web pages from the computer science departments of various universities, including the University of Texas and the University of Wisconsin. The dataset is commonly used for tasks such as webpage classification and link prediction, serving as a benchmark for evaluating machine learning models in graph-based learning scenarios. **Minesweeper** (Baranovskiy et al., 2023) dataset is a synthetic graph dataset inspired by the Minesweeper game. In this dataset, the graph is structured as a regular 100x100 grid, where each node represents a cell connected to its neighboring nodes, except for edge nodes, which have fewer neighbors. The primary task is to predict which nodes contain mines. This dataset is commonly used to evaluate the performance of GNNs under heterophily.

**Evaluation Metric.** Following mainstream FGL research experimantal practices, we utilize the accuracy of the node classification task as the evaluation metric.

**Baselines.** We compare $S^2$FGL with several state-of-the-art approaches, including traditional federated learning methods such as **FedAvg** (McMahan et al., 2017), **Fed-Prox** (Li et al., 2020), **FedNova** (Wang et al., 2020), and **FedFa** (Zhou & Konukoglu, 2023); federated graph learning approaches including **FGSSL** (Huang et al., 2023a) and **FGGP** (Wan et al., 2024a); as well as personalized federated graph learning methods such as **FedSage+** (Zhang et al., 2021b), **FedStar** (Tan et al., 2023), **FedPub** (Baek et al., 2023), and **FedGTA** (Li et al., 2024). This comprehensive set of baseline methods spans various FL and FGL paradigms, allowing us to evaluate the performance and advantages of our proposed $S^2$FGL across diverse scenarios.

**Implement Details.** Following prevalent methodologies in FGL research, we employ the Louvain community detection

algorithm to partition the graph into subgraphs assigned to different clients. For each dataset, we divide the nodes into training, validation, and testing sets with ratios of 60%, 20%, and 20%, respectively. Additionally, we simulate various collaborative scenarios by configuring the number of clients to 10 for Cora, Citeseer, Pubmed, and Minesweeper datasets, and 3 for the Texas and Wisconsin datasets. The primary evaluation metric is the node classification accuracy on the clients' test sets. We conduct each experiment five times and report the average accuracy from the last five communication epochs as the final performance. We conduct experiments with the ACM-GCN (Luan et al., 2022), which achieves a strong ability on both homophilic and heterophilic graph datasets.

### 5.2. Experiment Results

In this section, we comprehensively evaluate the proposed $S^2$FGL by addressing the following questions:

- **Q1: How does $S^2$FGL perform compared to existing FL and FGL methods in subgraph-FL?**

- **Q2: What is the impact of each component of $S^2$FGL on the overall performance?**

- **Q3: Does $S^2$FGL maintain consistent performance across different hyperparameter and client scales?**

- **Q4: Do NLIR and FGMA mitigate the effects of label signal disruption and spectral heterogeneity?**

**Q1: How does $S^2$FGL perform compared to existing FL and FGL methods in subgraph-FL?**

We present the results of node classification tasks across various FGL scenarios using multiple graph datasets, and

we summarize the final average test accuracy in Tab. 1. It demonstrates that our proposed $S^2$FGL consistently outperforms all baseline approaches across all six datasets. This superiority highlights the effectiveness of $S^2$FGL.

## Q2: What is the impact of each component of $S^2$FGL on the overall performance?

To evaluate the individual contributions of NLIR and FGMA strategies within the $S^2$FGL framework, we conducted ablation experiments on the Cora and Citeseer datasets. In this study, we removed each component to assess its impact on the overall performance. The results are presented in Tab. 2, which demonstrate that both NLIR and FGMA independently contribute to the overall performance.

| NLIR | FGMA | Dataset | |
| --- | --- | --- | --- |
| | | Cora | Citeseer |
| ✗ | ✗ | $81.9 \pm 0.7$ | $74.3 \pm 0.4$ |
| ✓ | ✗ | $83.2 \pm 0.4$ | $75.6 \pm 0.3$ |
| ✗ | ✓ | $82.6 \pm 0.3$ | $75.0 \pm 0.2$ |
| ✓ | ✓ | $\mathbf{83.4 \pm 0.2}$ | $\mathbf{76.0 \pm 0.3}$ |

*Table 2.* **Ablation study** on key components of $S^2$FGL.

## Q3: Does $S^2$FGL maintain consistent performance across different hyperparameter and client scales?

We evaluated the stability and adaptability of our proposed framework $S^2$FGL under varying hyperparameter configurations of and different client scales.

**Varying Hyperparameters of NLIR and FGMA.** For NLIR, we test hyperparameter sets of 100, 50, 10, and 1. For FGMA, the settings are 0.01, 0.05, 0.5, and 1. The results in Fig. 4 indicate that our method maintains consistent performance across varying hyperparameter configurations.

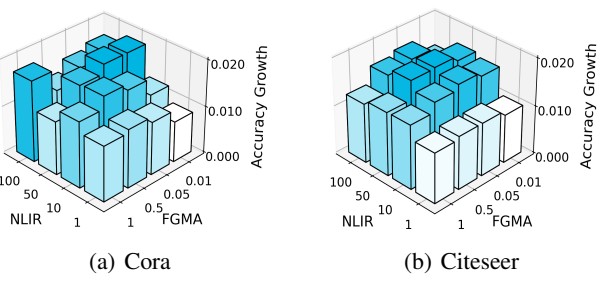

(a) Cora  (b) Citeseer

*Figure 4.* Analysis of the performance growth between $S^2$FGL and FedAvg under different hyperparameters of NLIR and FGMA

**Varying Client Scales.** We assessed performance with different client scales: 5, 10 and 20. Specifically, we compared $S^2$FGL with other FL and FGL baselines, including FedAvg, FedProx, and FGSSL. The results in Fig. 5 demonstrate that $S^2$FGL consistently delivers reliable results regardless of the client scales. Overall, $S^2$FGL exhibits strong stability

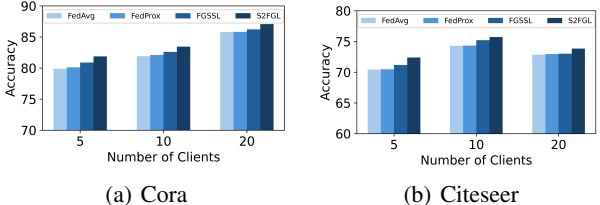

(a) Cora  (b) Citeseer

*Figure 5.* Analysis of performance under different client scales.

and adaptability across varying hyperparameter settings and client partition configurations on both the Cora and Citeseer datasets. These findings confirm that $S^2$FGL not only sustains its effectiveness under diverse conditions but also adapts seamlessly to varying client scales, demonstrating its suitability for real-world subgraph-FL scenarios.

## Q4: Do NLIR and FGMA mitigate the effects of label signal disruption and spectral heterogeneity?

In Fig. 1, our experiments demonstrate that the SIS score in subgraph scenarios declines compared to centralized training, while spectral heterogeneity exists among clients. Here, we further verify the targeted effectiveness of NILR and FGMA with respect to these two issues, respectively. First, in Fig. 6 (a), we investigate the relationship between the performance gain brought by NILR to FedAvg and the change in the SIS score. The results show that this method achieves higher performance when semantic signals are more limited, thereby confirming its effectiveness and targeted nature. Second, in Fig. 6 (b), we examine the relationship between the performance improvement of FGMA for FedAvg and spectral heterogeneity, measured by the average KL divergence between the eigenvalue distributions of different clients. Experimental results show that greater spectral heterogeneity corresponds to larger performance gains, confirming the effectiveness of our method. Specifically, green indicates the performance gain of the proposed method relative to FedAvg, blue denotes variations in SIS, and orange corresponds to spectral heterogeneity.

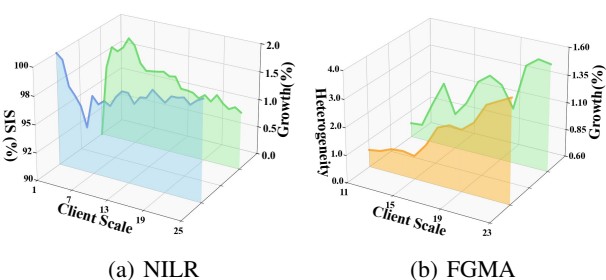

(a) NILR  (b) FGMA

*Figure 6.* Analysis of the targeted effectiveness of NILR and FGMA. (a) The performance gain from NILR increases as the semantic signal becomes limited. (b) The performance improvement from FGMA grows with higher spectral heterogeneity.

# 6. Conclusion

In this paper, we identify and empirically demonstrate two phenomena in subgraph-FL from both spatial and spectral perspectives of graph signal propagation: label signal disruption and spectral heterogeneity. These phenomena pose challenges of poor semantic knowledge and spectral client drift. To address these challenges, we propose two key strategies: NLIR and FGMA. NLIR selects structurally and semantically representative nodes and constructs a global repository accordingly. By injecting semantic information from the repository into local training, it alleviates the poor semantic knowledge caused by label signal disruption. In addition, FGMA aligns and feature projections in both the high- and low-frequency reconstructed graph spectra, thereby promoting a generic signal propagation paradigm and mitigating client drifts under spectral heterogeneity. By integrating these strategies, $S^2$FGL effectively tackles both spatial and spectral challenges in subgraph-FL. Extensive experiments on multiple datasets demonstrate that $S^2$FGL significantly enhances global generalizability.

# Acknowledgement

This work is supported by the National Key Research and Development Program of China (2024YFC3308400), and National Natural Science Foundation of China under Grant (62361166629, 62176188, 62225113, 623B2080), the Wuhan University Undergraduate Innovation Research Fund Project. The supercomputing system at the Supercomputing Center of Wuhan University supported the numerical calculations in this paper.

# Impact Statement

This paper presents work whose goal is to advance the field of Machine Learning. There are many potential societal consequences of our work, none of which we feel must be specifically highlighted here.

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
