# OpenReview forum: "$S^2$FGL: Spatial Spectral Federated Graph Learning"
_ICML.cc/2025/Conference — ICML 2025 poster_

### Official Review · Reviewer_uVfb · 2025-03-01

**Overall Recommendation:** 4

**Summary:**

This paper identifies and defines two significant limitations in FGL: label signal disruption and spectral client drifts. The proposed framework, S2S^2S2FGL, addresses these issues by consolidating globally accessible semantic knowledge and aligning the high- and low-frequency spectral domains. Abundant experiments validate its effectiveness.

**Claims And Evidence:**

The claims in this paper do not exhibit any significant flaws.

**Essential References Not Discussed:**

None.

**Experimental Designs Or Analyses:**

The experimental designs presented in this paper are reasonable. Specifically, the authors employ the Louvain partition, a widely used method for simulating real-world subgraph-FL scenarios, and utilize a standard node classification task as the evaluation criterion.

**Methods And Evaluation Criteria:**

The two methods effectively address the issues outlined in Figures 1 and 2. Additionally, the Louvain partition and the datasets used in the experiments are standard, making them appropriate for evaluating the effectiveness of these methods.

**Other Comments Or Suggestions:**

I have no additional comments or suggestions.

**Other Strengths And Weaknesses:**

Strengths：
The problems explored in this paper are intriguing. Drawing inspiration from graph active learning, the authors introduce the structure inertia score to measure the significance of semantic signal strength, highlighting a key challenge in subgraph-FL. This work also investigates the heterogeneity of spectral signal propagation across clients. Furthermore, the paper comprehensively addresses these issues in both domains and proposes reasonable solutions.

Weaknesses：
1. The discussion of current FGL research from lines 26 to 30 is too general. A more detailed analysis of existing methods and their limitations is required, particularly for the FGL baselines.
2. For the second method proposed in this paper, why does it align only the high-frequency and low-frequency spectral features, rather than other components? A more rational explanation is needed.

**Questions For Authors:**

Please see weaknesses.

**Relation To Broader Scientific Literature:**

The two issues identified in this paper are highly relevant and provide new insights for future FGL research.

**Theoretical Claims:**

NA

---

> ### Author Rebuttal · Authors · 2025-03-31
>
> Dear Reviewer uVfb:
>
> Thank you for your encouraging comments on our work. We hope that our responses below will address your concerns and reinforce your positive evaluation.
>
> ## Weakness
>
> **W1: A more detailed analysis of existing methods and their limitations
> is required, particularly for the FGL baselines.**
>
> A1: In response to your concerns, we further elaborate on the deficiencies of the FGL baseline methods evaluated in our study. Specifically, FedSage+ seeks to reconstruct subgraphs by locally training a neighborhood generator. FedGTA determines aggregation weights using mixed moments of neighbor features and local smoothing confidence. FGSSL aligns similarity matrices between local and global GNNs while employing node-level contrastive learning. Similarly, FGGP applies contrastive learning, integrating clustering prototypes to enhance prediction accuracy. FedPub utilizes similarity-based clustering on random graphs at the client side and employs local masks to incorporate the global model. FedStar introduces a specialized channel for processing graph structure, sharing only this channel’s parameters. In summary, existing approaches fail to address two critical issues: spatial label signal disruptions and spectral client drifts. Label signal disruptions weaken the semantic recognition ability of GNNs in FGL, thereby reducing its overall effectiveness. Additionally, spectral client drifts hinder collaboration among clients, making it difficult to establish an efficient spectral signal propagation paradigm. Our work aims to address this gap by focusing on these two underexplored aspects.
>
> **W2: Why does the second method align only the high-frequency and low-frequency spectral features, rather than other components?**
>
> A2: Both low-frequency and high-frequency areas of the graph spectrum are critical and highly representative. Low-frequency components typically capture the global structural properties of the graph, for instance, the second smallest eigenvalue indicates graph connectivity. Conversely, high-frequency eigenvectors contain detailed local information. Directly matching all eigenvectors is computationally prohibitive. Instead, selecting those associated with the smallest and largest eigenvalues reduces computational complexity while preserving the graph's essential structural characteristics.
>
>
> We are grateful for your encouraging feedback and hope that our response has effectively addressed your concerns!

---

> > ### Comment · Reviewer_uVfb · 2025-04-02
> >
> > I appreciate the thorough rebuttal you have provided and will maintain my current score.
> >
> > Regarding the comments from other reviewers, I would like to provide some additional insights.
> >
> > While edge loss is acknowledged in FGL, its semantic impact remains unexplored. This paper examines label signal disruption and quantifies the relationship between client scale and semantic degradation, offering new insights. Moreover, the moderate complexity of the proposed method is reasonable, especially given its strong performance.
> >
> > Overall, I recommend its acceptance.

---

### Official Review · Reviewer_yo13 · 2025-03-04

**Overall Recommendation:** 4

**Summary:**

The authors propose S2FGL, a framework that simultaneously addresses the spatial and spectral challenges in federated graph learning. Instead of focusing on static graph structures, it provides solutions through the lens of graph signal propagation.

**Claims And Evidence:**

Given the inherent interconnection between the spatial and spectral domains, methods that perform well in the spatial domain are expected to positively impact the spectral domain. However, the authors do not adequately explain the key benefits of utilizing the spectral domain to address spectral client drifts.

**Essential References Not Discussed:**

None

**Experimental Designs Or Analyses:**

The experiments are comprehensive. However, the authors should consider the impact of the number of representative prototypes for each class in the NLIR on performance and conduct a hyperparameter study. Furthermore, the performance comparison experiment should include FedTAD [1].

[1] FedTAD: topology-aware data-free knowledge distillation for subgraph federated learning. IJCAI 2024

**Methods And Evaluation Criteria:**

The solutions provided by S2FGL are targeted, and the datasets used are commonly employed in federated graph learning research.

**Other Comments Or Suggestions:**

There is a missing punctuation mark after the equation (9).

**Other Strengths And Weaknesses:**

The primary strengths and weaknesses have been discussed above.

**Questions For Authors:**

1. The authors should conduct a hyperparameter study for the NLIR and include a performance comparison with FedTAD.
2. The authors should clarify the key advantages of the spectral solutions over spatial solutions in mitigating spectral client drifts.

**Relation To Broader Scientific Literature:**

The problems explored in this study are meaningful. First, the authors shift the focus from the typical issue of heterogeneity and examine the shortcomings of FGL in comparison to centralized graph learning from a spatial perspective. Second, the spectral domain has received insufficient attention in existing FGL studies, and this paper contributes to filling that gap.

**Theoretical Claims:**

The definitions are rigorous

---

> ### Author Rebuttal · Authors · 2025-03-31
>
> Dear Reviewer yo13:
>
> Thank you for your thoughtful review and for acknowledging the value of our work. We hope the responses provided below will clarify your concerns and contribute to a more favorable evaluation.
>
> ## Weakness
>
> **W1: The authors should conduct a hyperparameter study for the NLIR and include the performance comparison with FedTAD.**
>
> A1: Thank you for your thoughtful suggestions. We have added experiments regarding the prototype count hyperparameters in NLIR to Table 1 and included comparison experiments with FedTAD in Table 2. The results in Table 1 demonstrate the stability of the NLIR method under variations in prototype counts. The results in Table 2 show that our method consistently outperforms FedTAD, validating its efficiency. We will include these results in the final version.
>
> *Table 1: Hyper-parameter study of prototype count in NLIR*
> | Prototype Count | Cora | Citeseer | Pubmed |
> |:-:|:-:|:-:|:-:|
> | 4  | 83.4 | 76.0 | 88.6 |
> | 8  | 83.6 | 76.3 | 88.5 |
> | 16 | 83.5 | 76.4 | 89.0 |
>
> *Table 2: Performance comparison with FedTAD*
> |Datasets|Cora|Citeseer|PubMed|Texas|Wisconsin|Minesweeper|
> |-|-|-|-|-|-|-|
> |FedAvg|81.9|74.3|87.3|72.8|77.6|79.6|
> |FedTAD|82.5|75.3|87.7|73.4|77.7|79.9|
> |$\bf{S^2FGL}$|**83.4**|**76.0**|**88.6**|**74.7**|**78.4**|**80.5**|
>
>
> **W2: The authors should clarify the key advantages of the spectral solutions over spatial solutions in mitigating spectral client drifts.**
>
> A2: Methods that rely on the spatial domain fail to capture signal propagation patterns across different frequencies in the graph spectra. As a result, client drifts caused by GNNs overfitting to local frequency signal propagation paradigms are challenging to resolve for them. Instead, our method performs spectral reconstruction based on the adjacency awareness of different GNNs and promotes the alignment of local spectral signal propagation patterns with the globe. This approach enables the formation of a generalizable and strong spectral signal propagation paradigm, effectively mitigating spectral client drifts.
>
>
> Thank you again for your efforts in reviewing our work. We hope our rebuttal has clarified and addressed your concerns!

---

### Official Review · Reviewer_yM48 · 2025-03-07

**Overall Recommendation:** 4

**Summary:**

This paper investigates graph signal propagation in federated graph learning through both the spatial and spectral domains, highlighting the issues of label signal disruption and spectral client drift. In response, it proposes two methods: Node Label Information Reinforcement and Frequency-aware Graph Modeling Alignment, which address these identified challenges. Comprehensive experiments are performed to demonstrate the effectiveness of these methods.

**Claims And Evidence:**

Claims are supported by clear and convincing evidence.

**Essential References Not Discussed:**

All essential references have been included, especially those related to the latest methods in subgraph-FL.

**Experimental Designs Or Analyses:**

This paper presents extensive experimental evidence, encompassing performance comparisons, ablation studies, and hyperparameter studies, among others. Overall, the experimental framework is well-constructed. Notably, in the Q4 of the experiments, the correlation between the NLIR method and the structure inertia score is examined, directly demonstrating the method's relevance. However, this raises a concern: under larger partition sizes, SIS does not significantly decrease. In such instances, will NLIR lose its original effectiveness? Consequently, the performance of NLIR with a substantial number of clients still needs to be validated, ideally with the SIS reduction level indicated. Furthermore, the ablation study should incorporate error bars to provide a more comprehensive analysis of the results.

**Methods And Evaluation Criteria:**

The proposed methods are well-designed and appropriately address the identified challenges. The NLIR method creates a global category knowledge repository by enabling clients to upload multiple easily-learnable prototypes as reference points. By evaluating the similarity between each node and these reference points, it achieves precise semantic localization of nodes, thereby integrating globally accessible category information into local GNNs. The FGMA method utilizes the similarity matrix obtained from GNN inference to perform spectral reconstruction, aligning local and global features by projecting them onto the low-frequency and high-frequency spectral domains. The reconstruction allows FGMA to synchronize local and global message passing paradigms, effectively alleviating spectral drift.

**Other Comments Or Suggestions:**

Under the Notations section, there is a typo in line 122 where a word is repeated.

**Other Strengths And Weaknesses:**

Strengths:
1. The motivation is both strong and innovative, with exploration of the issues of label signal disruption and spectral client drift.
2. The proposed solutions are well-suited and specifically targeted, effectively addressing the identified problems in the spatial and spectral domains.
3. The experimental design is generally rigorous and effective. Moreover, the experiment in Q4, which examines the relationship between the performance of the NLIR method and SIS, successfully validates the motivation..

Weaknesses:
1. In the experiment Q4, as the partition size increases, SIS rises, leading to a decline in the performance of the NLIR. The authors should consider conducting experiments with a larger number of clients to ensure that NLIR remains effective in configurations with more clients.
2. The ablation study should incorporate error bars for enhanced clarity and precision in the results.

**Questions For Authors:**

Please refer to the weaknesses, as no other questions have been posed.

**Relation To Broader Scientific Literature:**

This paper innovatively identifies the issue of label signal disruption , with a strong motivation that is empirically validated. Moreover, the investigation into spectral drift presents a novel perspective in the context of subgraph-FL, encouraging further exploration of graph spectral domain in FGL.

**Theoretical Claims:**

The definitions are clear and correct, and there is no theoretical proof provided.

---

> ### Author Rebuttal · Authors · 2025-03-31
>
> Dear Reviewer yM48:
>
> We sincerely appreciate the time and effort you have invested in reviewing our paper, as well as your favorable assessment of the motivation and design of our method. We hope that our rebuttal has effectively addressed your concerns.
>
>
> ## Weakness
>
> **W1: The authors should consider conducting experiments with a larger number of clients to ensure that NLIR remains effective in configurations with more clients**
>
> A1: We kindly emphasize that our experimental setup adheres to standards in existing FGL research, incorporating suitably configured numbers of clients. Nonetheless, to address your concerns, we conducted additional experiments with a substantially larger number of clients on the large-scale graph dataset arxiv-year. The results in Table 1 confirm the robustness and effectiveness of our method in large-scale client settings.
>
>
> *Table 1: NLIR performance under large client scale.*
>
> | Client Scale | 50 | 100 | 150 |
> |:-:|:-:|:-:|:-:|
> | FedAvg  | 32.1  | 34.8 | 35.5 |
> | NLIR  | 32.9  | 35.7 | 36.2 |
>
>
> **W2: The ablation study should incorporate error bars.**
>
> A2: In response to your concern, we incorporate error bars in Table 2 as follows. We will incorporate your suggestions in the final version.
>
>
> *Table 2: Ablation study of $S^2FGL$ with error bars*
>
> | NLIR | FGMA | Cora | Citeseer |
> |:-:|:-:|:-:|:-:|
> | ✗ | ✗ | 81.9 ± 0.7     | 74.3 ± 0.4     |
> | ✓  | ✗  | 83.2 ± 0.4     | 75.6 ± 0.3     |
> | ✗  | ✓ | 82.6 ± 0.3     | 75.0 ± 0.2     |
> | ✓ | ✓  | **83.4 ± 0.5** | **76.0 ± 0.3** |
>
> ## Other Comments Or Suggestions
>
> A3: We are grateful for your feedback regarding the typo. We will implement the revision in the final version.
>
>
> We sincerely appreciate your thoughtful review comments and hope that our rebuttal has addressed the concerns you raised!

---

### Official Review · Reviewer_svqb · 2025-03-07

**Overall Recommendation:** 3

**Summary:**

The paper presents a novel framework called S2FGL (Spatial Spectral Federated Graph Learning) to address two key challenges in subgraph federated learning (FGL): Label Signal Disruption (LSD) and spectral client drifts. LSD occurs when subgraphs lose critical label signals due to edge losses between clients, which hampers the ability of Graph Neural Networks (GNNs) to learn class knowledge. Spectral client drifts arise from inconsistencies in signal frequencies across subgraphs, leading to degraded global generalizability. To solve these issues, the authors propose two strategies: Node Label Information Reinforcement (NLIR), which creates a global repository of class knowledge to restore label signals, and Frequency-aware Graph Modeling Alignment (FGMA), which aligns high- and low-frequency spectral components across clients to mitigate spectral drifts. Extensive experiments on various datasets demonstrate the effectiveness of S2FGL, outperforming existing methods in terms of global generalizability.

**Claims And Evidence:**

The label signal disruption in subgraph-FL has already been recogonized and can not be taken as one contribution of this work.

**Essential References Not Discussed:**

No

**Experimental Designs Or Analyses:**

There are only experiments with one strong GNN backbone, ACM-GCN. But many graph federated learning works use other more naive but popular GNNs, e.g., GCN and GPRGNN. The authors should include the experiments with other GNNs used in prior works.

**Methods And Evaluation Criteria:**

Yes

**Other Comments Or Suggestions:**

see Above

**Other Strengths And Weaknesses:**

**Strengths:**

- This paper is well-organized and easy to follow.

- Although the spectral client drift problem is quite intuitive to conceive, this paper appears to be the first to explicitly pinpoint this issue.


**Weakness:**

- The problem NLIR aims to address and NLIR itself are not new things in federated learning since there are many works discussing the loss of egdes and prototype-based federeated learning. The authors should emphasize the innovative aspects of their method in addressing Challenge , compared with prior works.

- The cosine similarity matrix requires second-order complexity. And the projection step on p. 6 requires doing EVD on a dense Laplacian.

- The Spectral Client Drift challenge is novel. But the authors do not experimentally verify the existence of the second challenge or whether their proposed method can mitigate the spectral domain distribution shift of graph signals. It would be beneficial for the authors to provide experimental validation for Challenge 2, rather than relying on mere conjecture.

- Complexity analysis is missing.

**Questions For Authors:**

see Above

**Relation To Broader Scientific Literature:**

This article addresses a common intra-graph federated learning task, identifying two challenges it faces: 1) Label Signal Disruption caused by the absence of cross-client links, and 2) Spectral Client Drift, a distribution shift resulting from inconsistent spectral domain distributions of graph signals across different clients. The first challenge is widely recognized, while the second has received little attention. Overally, the proposed method itself is a combination of existing techniques.

**Theoretical Claims:**

There are no theoretical proofs.

---

> ### Author Rebuttal · Authors · 2025-03-31
>
> Dear Reviewer svqb:
>
> We sincerely appreciate your time and effort and hope that our responses will address your concerns and lead to an updated score.
>
> ## Experimental Designs Or Analyses
> **There are only experiments with ACM-GCN.**
>
> A1: $S^2FGL$ is a backbone-agnostic framework designed to address label signal disruption and spectral client drifts. In light of your thoughtful comment, we have incorporated results using GCN in Table 1, showing that $S^2FGL$ consistently outperforms current methods. The results will be included in the final version.
>
> *Table 1: Performance comparison using **GCN.***
> |Method|Cora|Citeseer|PubMed|Texas|Wisconsin|Minesweeper|
> |-|-|-|-|-|-|-|
> |FedAvg|80.2|69.5|84.9|63.4|62.1|77.9|
> |FedProx|80.3|69.2|85.2|64.0|62.2|78.0|
> |FedFA|80.8|70.1|85.4|64.7|63.4|78.3|
> |FedSSL|82.6|70.9|85.8|64.2|64.5|78.6|
> |FGGP|82.8|70.6|85.5|65.5|63.6|78.7|
> |$\bf{S^2FGL}$|**83.7**|**72.2**|**86.8**|**66.8**|**65.5**|**79.1**|
>
> ## Weakness
> **W1: Why NLIR is novel.**
>
> A2: For the target problem, we innovatively focus on label signal disruption (LSD), which brings significant semantic degradation. Though existing methods mitigate edge loss structurally, they ignore the semantic degradation under LSD and fail to offer targeted solutions. Therefore, current FGL methods inevitably suffer from limited GNN semantic recognition ability.
>
> Methodologically, existing prototype-based approaches exhibit two critical limitations under LSD: insufficient semantic richness and semantic deviation under structural biases. Correspondingly, we propose a novel metric $\Lambda^\text{SALC}$ (Eq. 3 and 4, Page 4), which jointly accounts for label influence and structural representativeness to effectively mitigate LSD.
>
> **W2: The similarity matrix requires second-order complexity. The projection requires doing EVD on a dense Laplacian.**
>
> A3: First, the computation of the feature similarity matrix is widely accepted in FGL. Specifically, FGSSL leverages it to align structural consensus, FGGP utilizes it for contrastive learning, and FedGL employs it for structure updates. Distinctively, we align the reconstructed spectrum to enable a highly expressive signal propagation scheme against spectral drifts, achieving the best performance.
>
> Second, our method **does not** require an EVD on a dense Laplacian. Since only partial eigenvectors are needed, we leverage SciPy's sparse solver eigsh with $O(r\cdot nnz\cdot I)$. $r$ denotes the number of eigenvectors needed, $nnz$ is the non-zero element, and $I$ is required iterations.
>
> **W3: The authors need to verify the existence of the spectral challenge and prove the method can mitigate it.**
>
> A4: We kindly note that we have demonstrated the spectral challenge in Fig. 1(b). For further validation, we compute the Pearson correlation between spectral shift (KL divergence of a client's eigenvalue distribution from the global) and local accuracy of the global GNN under FedAvg. After runs under five random seeds, they are -0.39 (Cora), -0.27 (Citeseer), and -0.34 (Pubmed), indicating that greater spectral shift leads to higher inconsistency with global optimization and confirming spectral drifts.
>
> FGMA facilitates a generic signal propagation paradigm across clients to mitigate spectral drifts. For validation, Table 2 shows that FGMA is **more** effective under **higher** spectral heterogeneity, providing strong evidence for its specificity and effectiveness. Spectral heterogeneity here is the average KL divergence among clients' eigenvalue distributions.
>
> *Table 2: Correlation between **Spectal Heterogeneity** and **FGMA Improvement** on FedAvg.*
> |Client Scale|11|13|15|17|19|21|23|
> |-|-|-|-|-|-|-|-|
> |Spectral Heterogeneity|0.63|0.88|0.85|2.09|2.23|3.30|3.75|
> |FGMA Improvement (%)|0.73|0.97|0.92|1.27|1.29|1.51|1.57|
>
> **W4: Lack of complexity analysis.**
>
> A5: For a $k$-layer GNN with batch size $b$ and feature dimension $f$, the propagated features $X^{(k)}$ have a space complexity of $O((b+k)f)$, while linear regression has $O(f²)$. Key parameters include $n$, $m$, $c$ (nodes, edges, classes), $s$ (augmented nodes), $g$ (complemented neighbors), $p$ (trainable prototype matrix dimension), $Q$ (query set size for contrastive learning), and $N$ (selected clients per round). Analysis in Table 3 shows that $S^2FGL$ achieves top performance with reasonable overhead.
>
> *Table 3: Complexity analysis. Best in bold and second with underline.*
> |Method|Client Mem|Server Mem|Client Time|Server Time|
> |-|-|-|-|-|
> |FedStar|$\bf{O(2((b+k)f+f^2))}$|$\underline{O(N f^2)}$|$\bf{O(2(kmf+nf^2))}$|$\bf{O(Nf)}$|
> |FGSSL|$O(Q(b+k)f+f^2+n^2)$|$\underline{O(Nf^2)}$|$O(Qkmf+Qnf^2+n^2f)$|$\bf{O(Nf)}$|
> |FGGP|$O((n+sg)f+f^2+Qcp+n^2)$|$\bf{O(Ncp)}$|$O((m+sg)f+(n+sg)f^2+Qcp^2)$|$\underline{O(N^2(\log(N)+c^2p^2)+Ncp)}$|
> |$S^2FGL$|$\underline{O((b+k)f+f^2+n^2)}$|$\underline{O(Nf^2)}$|$\underline{O(kmf+nf^2+n^2f)}$|$\bf{O(Nf)}$|
>
> Thank you for your valuable feedback and hope that our rebuttal adequately addresses your concerns!

---

### Decision · Program_Chairs · 2025-05-01

**Decision:**

Accept (poster)

**Comment:**

The reviewers highlighted this paper's novel setting, good writing, and convincing experiments. The authors also provided reasonable answers to the doubts about the details of the proposed dataset and method. The reviewers’ concerns were well-addressed. Therefore, the decision is to recommend Acceptance.